# OMNICLEAR: SOFT EFFECTS REMOVAL FROM IMAGES WITHIN A VERSATILE MODEL

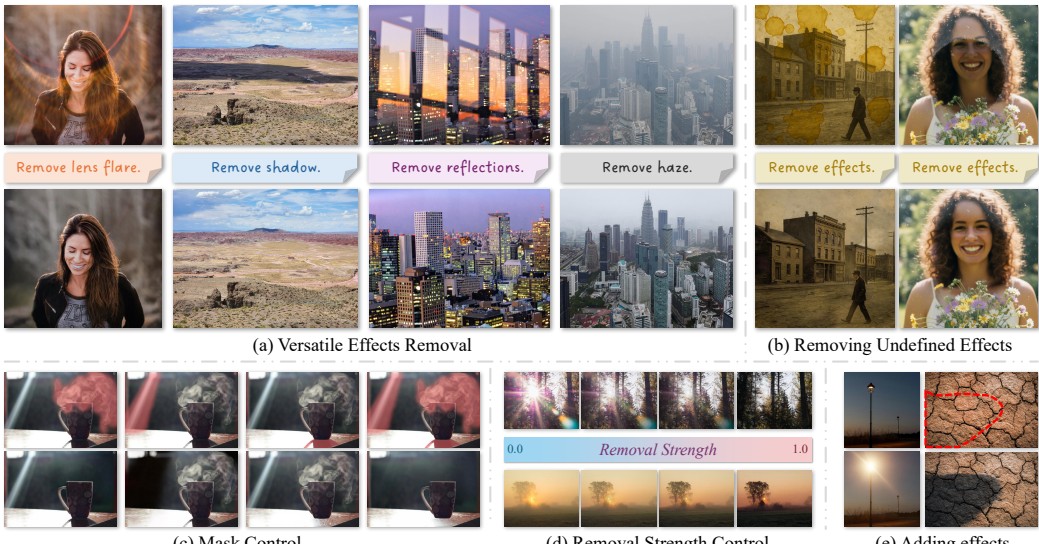

Figure 1: Our OmniClear eliminates multiple challenging (a) and even undefined (b) soft effects from in-the-wild images while preserving background identities. Besides, OmniClear supports precise pixel mask control (c), and removal strength control (d), allowing for intuitive and fine-grained restoration tailored to specific user needs. The framework is also capable of adding effects in the given region (e). Masks are global by default if not shown. **A demo video is included in the supplementary materials.**

## ABSTRACT

Digital images are often degraded by soft effects such as lens flare, haze, shadows, and reflections, which reduce aesthetics even though the underlying pixels remain partially visible. The prevailing works address these degradations in isolation, developing highly specialized, specialist models that lack scalability and fail to exploit the shared underlying essences of these restoration problems. While specialist models are limited, recent large-scale pretrained generalist models offer powerful, text-driven image editing capabilities. while recent general-purpose systems (*e.g.*, GPT-4o, Flux Kontext, Nano Banana) require detailed prompts and often fail to achieve robust removal on these fine-grained tasks or preserve identity of the scene. Leveraging the common essence of soft effects, *i.e.*, semi-transparent occlusions, we introduce a foundational versatile model, capable of addressing diverse degradations caused by soft effects within a single framework. Our approach centers on fine-tuning a potent inpainting model on a large-scale, curated dataset of paired images, enabling it to learn robust restoration priors. Our method provides simple and intuitive user control, either global removal or mask-based removal with strength control, making interaction easier while ensuring higher reliability. Extensive experiments demonstrate that our unified model outperforms both prior specialist methods and popular general-purpose models, achieving robust and stable performance on in-the-wild scenarios.

## 1 INTRODUCTION

Images captured in real-world environments inevitably suffer from degradations. A common class of such "soft" effects includes optical phenomena (e.g., lens flare, reflections) and atmospheric conditions (e.g., haze, fog). These effects corrupt scene radiance additively or multiplicatively, degrading contrast, color fidelity, and fine details (Le & Samaras, 2019; Wan et al., 2017). Consequently, image quality and visibility are compromised, and in severe cases, occlusions cause irreversible information loss, rendering recovery fundamentally ill-posed (He et al., 2010; Wu et al., 2021; Le & Samaras, 2019).

To restore image structures, most existing works address each degradation type separately. For instance, dehazing has progressed from prior-based methods such as the Dark Channel Prior (DCP) (He et al., 2010) to deep networks estimating scattering parameters or directly predicting clean images (Li et al., 2017; Song et al., 2023; Chen et al., 2021; Engin et al., 2018; Chen et al., 2019). Similarly, shadow, flare, and reflection removal adopt task-specific designs (Le & Samaras, 2019; Dong et al., 2024; Wu et al., 2021; Xue et al., 2025; Zhu et al., 2024; Wan et al., 2017), relying on physical modeling, layer decomposition, or elaborate data and network strategies to mitigate ill-posedness. While such methods achieve strong task-specific performance, recent works (Chen et al., 2025a; Li et al., 2020b; Potlapalli et al., 2023) attempt to unify multiple degradations within one framework. Yet these models remain limited in scalability and robustness when facing extreme, diverse real-world conditions. This motivates the development of foundation models trained on large-scale data to achieve stronger generalization and resilience in the wild.

Concurrently, the rise of powerful foundation models like GPT-4o (Hurst et al., 2024) and Nano Banana (Gemini 2.5 Flash Image) (Comanici et al., 2025; Google, 2025) has introduced general-purpose, text-driven image generation/editing based on Multi-modal Large Language Models (MLLMs). These models can interpret complex prompts and perform realistic edits. However, for fine-grained tasks like soft effect removal, they exhibit significant limitations. Their performance is often unstable and heavily reliant on meticulously crafted text prompts. More critically, they lack the precise, pixel-wise control required for high-fidelity restoration and identity preservation. Treating soft effect removal as a general inpainting task can lead to the alteration of local image structures or the identity of objects within the scene, which makes them unreliable for professional photo editing and critical computer vision pipelines.

Despite their diverse appearances, effects such as lens flare, haze, reflections, and shadows share the same intrinsic property: they are all semi-transparent occlusions that preserve the identity of the underlying scenes. To this end, we define a unified and extensible task, termed Soft Effects Removal (SER). This task is highly challenging. First, these effects are typically entangled with the scene itself, rather than merely superimposed as simple overlays. Second, the local image structures, and even pixel-level identities, should be precisely preserved. Third, regions that are fully occluded or invisible (*e.g.*, overexposed areas in lens flare or areas covered by extremely dense haze) must be plausibly reconstructed.

To effectively tackle these challenges, we introduce **OmniClear** (Fig. 1 (a) & (b)), a data-centric versatile model for Soft Effects Removal. Our method is built upon two key points. First, we curated a large-scale dataset of approximately 3.8M balanced, high-quality, pixel-aligned image pairs. By unifying existing open-source datasets and augmenting them with extra real-world and synthetic data, we provide the precise supervision our model needs to learn content invariance. Second, as shown in Fig. 1 (c) & (d), we implemented fine-grained user controls, including pixel-level masks to define the removal area and strength levels to modulate the removal strength, making the process highly controllable. Beyond restoration, OmniClear can also perform aesthetic edits, such as enhancing existing effects or generating new, realistic ones on clean images (Fig. 1 (e)). Our method achieves state-of-the-art results on multiple public benchmarks and demonstrates significantly better generalization on in-the-wild testing data.

In summary, our main contributions can be summarized as follows:

- **A Versatile SER Model:** Proposed a single, versatile model OminClear for removing diverse soft effects in the wild. Our model achieves state-of-the-art performance on each task and surpasses much larger general-purpose models such as Nano Banana.
- **A Large-Scale Dataset for Generalization:** Curated a large-scale dataset of∼3.8M image pairs, providing vast data distribution for strong generalization on challenging in-the-wild data.

- **Controllable Editing:** Developed fine-grained user controls for SER tasks, including spatial masks and strength levels, to enable precise and controllable effect removal.

## 2 RELATED WORK

### 2.1 ISOLATED EFFECTS REMOVAL

**Lens flare removal.** Previous learning-based methods improved data synthesis by considering camera ISP to enhance realism and generalization (Zhou et al., 2023; 2025). Concurrently, architectural innovations emerged, including self-supervised methods to disentangle co-occurring flares (He et al., 2025), while others explicitly separated light source preservation from flare removal using dedicated detection modules (Ghodesawar et al., 2023), and networks leveraging both spatial and frequency domains (Vasluianu et al., 2024). More recently, large pretrained Latent Diffusion Models (LDMs) are adapted to leverage their powerful generative priors (Zhou et al., 2024). The development of these methods has also been heavily reliant on specialized datasets, from semi-synthetic ones (Wu et al., 2021), Flare7K (Dai et al., 2022), to real-world paired datasets (Dai et al., 2024).

**Reflection removal.** Early methods for single-image reflection removal (SIRR) focused on iterative refinement using edge maps (Fan et al., 2017) or recurrent networks (Yang et al., 2018; Li et al., 2020a). Subsequent research shifted towards improving training data realism by learning non-linear blending (Wen et al., 2019), employing physically-based rendering (Kim et al., 2020), and modeling glass absorption (Zheng et al., 2021). Architectural innovations followed, introducing location-aware modules (Dong et al., 2021) and advanced attention mechanisms (Huang et al., 2025; Zhang et al., 2025) to better distinguish between layers. More recent paradigms reduce reliance on paired data through unsupervised deep image priors (RahmaniKhezri et al., 2022) or by using Diffusion Models to generate guiding prompts (Wang et al., 2024a). This progress has been underpinned by the creation of key real-world benchmarks like $SIR^2$ (Wan et al., 2017) and the large-scale RRW dataset (Zhu et al., 2024).

**Shadow removal.** Initial approaches to shadow removal relied on traditional physical priors and optimization frameworks (Guo et al., 2011; Zhang et al., 2015). The advent of deep learning introduced end-to-end models like DeshadowNet (Qu et al., 2017) and methods that decomposed images into shadow-free and matte layers (Le & Samaras, 2019). Subsequent architectural advancements included using Generative Adversarial Networks (GANs) for joint detection and removal (Wang et al., 2018), fusing synthetic exposure pairs (Fu et al., 2021), and learning via shadow generation (Liu et al., 2021). More recent trends focus on eliminating the dependency on explicit shadow masks, utilizing mask-free transformers (Dong et al., 2024) or reformulating the problem as a dense prediction task (Lin et al., 2025). The progress in this field has been propelled by benchmarks like SRD (Qu et al., 2017), ISTD (Wang et al., 2018), and the newer high-resolution WSRD dataset (Vasluianu et al., 2023).

**Haze removal.** Single-image dehazing evolved from early methods based on statistical priors like the Dark Channel Prior (DCP) (He et al., 2010) to data-driven deep learning. Initial deep learning works included lightweight end-to-end networks (Li et al., 2017), hybrid models that learned priors for traditional optimization (Yang & Sun, 2018), and unpaired training with GANs to address data scarcity (Engin et al., 2018). Architectural innovations, such as gated context aggregation (Chen et al., 2019) and Vision Transformers (Song et al., 2023), were later introduced to better handle non-uniform haze. Recent efforts focus on closing the synthetic-to-real domain gap by generating more physically plausible training data (Chen et al., 2021) or leveraging diffusion models for realistic haze synthesis (Wang et al., 2025). This progress has been consistently driven by the development of comprehensive benchmarks (Li et al., 2018; Zhang et al., 2024; Islam et al., 2024).

Apart from them, some works delve into unified methods to restore image quality from multiple degradations caused by bad weathers (Li et al., 2020b; Potlapalli et al., 2023; Chen et al., 2025a). Despite the achievements from all these methods, key challenges persist including the domain gaps of synthetic datasets and limited diversity in real-world datasets, while current methods still struggle with scalable training with robust generalization abilities, as well as handling various types of challenging soft effects simultaneously.

### 2.2 PROMPT-BASED IMAGE EDITING

Prompt-based image editing originated from diffusion models, enabled by deterministic inversion techniques like DDIM (Song et al., 2020) that map real images to an editable latent space. Initial methods controlled edits by manipulating internal model structures, such as altering cross-attention

maps to preserve layout (Hertz et al., 2022) or fine-tuning the entire model on a single image for complex, non-rigid changes (Kawar et al., 2023). The field has since evolved towards more direct user control, with models trained to follow natural language instructions (Brooks et al., 2023) or allow for interactive, point-based spatial adjustments (Shi et al., 2024). This shift towards more precise, semantic editing is increasingly powered by the advanced contextual understanding of Multimodal Large Language Models (MLLMs) (Hurst et al., 2024; Comanici et al., 2025; Bai et al., 2025). However, current approaches still often lack fine-grained pixel control and can struggle to perfectly preserve the subject's identity during transformation.

## 3 METHODOLOGY

### 3.1 DATA CURATION

A powerful foundation model requires large-scale, high-quality, and diverse training data. To equip OmniClear with robust generalization, we curated a comprehensive dataset by unifying pixel-aligned image pairs from four representative tasks: lens flare, shadow, haze, and reflection removal. This integration enables the model to learn a broad restoration representation while preserving content identity.

**Public datasets.** We incorporate multiple benchmark datasets spanning the four domains (see Table 1 and supplementary materials for details). Despite their usefulness, these datasets exhibit imbalance, such as the scarcity of large-scale flare removal data and limited diversity in haze scenarios.

**Data expansion.** To remedy these gaps and increase data volume, we expand training data through three complementary sources: real-world captures, 2D synthesis, and 3D rendering.

- *Lens flare.* The key bottleneck lies in insufficient data. We therefore construct 78 indoor and outdoor 3D scenes in Blender (Blender Online Community, 2018), rendering about 70K paired images, named HALO dataset. Unlike Flare7K (Dai et al., 2022), which overlays flare layers on clean images, our rendered data produce physically consistent and realistic flare effects. The dataset covers diverse flare patterns, including reflective flare, glare, shimmer, and streaks.
- *Shadow.* While public datasets cover both indoor and outdoor scenes, they contain only ∼5K pairs. To scale up, we add an additional 26K photo pairs. Specifically, we repurpose internal object-effect removal data: by stitching objects without shadows into background images, we synthesize corresponding shadow-free versions to form the Large Real-world Shadow Removal Dataset (LR-SRD).
- *Haze.* Existing synthetic datasets (RESIDE, HAZESPACE) often appear uniform or algorithmically simplistic. To generate more realistic and challenging cases, we use their clean ground-truth images with monocular depth (Ke et al., 2025), and apply a physically motivated atmospheric rendering pipeline. This allows precise control of parameters such as visibility, airlight color, scatter, and optical thickness. To simulate non-homogeneous haze or fog, we introduce procedural noise fields and path blurring, yielding realistic textures of haze, smoke, and fog. More synthesis details are provided in the supplementary material.

These expanded datasets extend coverage to underrepresented scenarios and complement public benchmarks, enhancing OmniClear's robustness in the wild. A detailed breakdown is given in Table 1, with representative samples in Fig. 2.

### 3.2 FRAMEWORK

As shown in Fig. 3, OmniClear is a unified framework designed to tackle multiple soft effect removal tasks. Inspired by UniReal (Chen et al., 2025b), the core architecture reformulates these diverse tasks as a problem of *discontinuous frame generation* within a latent diffusion model. The process begins with a Variational Autoencoder (VAE) (Kingma & Welling, 2013) encoding the input image into a compact latent space, while a text encoder processes a task-specific prompt (*e.g.*, "*remove haze*") to generate instructive embeddings. These conditional inputs (image latent and textual embeddings) are then concatenated with the noisy target latent and fed as a sequence to a Diffusion Transformer (DiT). The DiT's full attention mechanism operates on this sequence, allowing it to iteratively predict and remove noise from the target latent by conditioning on both the visual context and the textual instructions. Finally, the fully denoised latent is passed through the VAE decoder to reconstruct the final, effect-free image. The model is trained using a mean squared error (MSE)

Table 1: Summary of datasets curated for OmniClear training. "†" represents the datasets curated by us, "*" represents the datasets which we re-synthesis effects with our own algorithm.

| Task | Dataset | Type | Description | Pairs |
|------|---------|------|-------------|-------|
| Lens flare | FlareReal600 (Dai et al., 2024) | Real-World | Nighttime flares, Streetview, Cityscapes, Outdoor | 0.6k |
| | HALO† | 3D Synthetic | Rendered, Various flares and scenes, Indoor & Outdoor | 70k |
| Shadow | WSRD+ (Vasluianu et al., 2023) | Real-World | Object-level, Close-view, Rich texture, Complex shadows | 1k |
| | ISTD+ (Wang et al., 2018) | Real-World | Simple-shaped shadows, Monotonous scenes, Outdoor | 1.3k |
| | SRD (Qu et al., 2017) | Real-World | Various scenes, Outdoor | 2.6k |
| | LR-SRD† | Real-World | Object-level, Close-view, Hard & Soft shadow, Indoor & Outdoor | 26k |
| Haze | Haze-R (Ancuti et al., 2018b;a; 2019; 2020; 2021; 2023; 2024) | Real-World | Collection includng: I-HAZE, O-HAZE, Dense-Haze, NH-Haze, etc., Homogeneous & Non-Homogeneous, Indoor & Outdoor | 0.3k |
| | REVIDE (Zhang et al., 2021) | Real-World | Video Frames, Indoor | 1.9k |
| | LM-Haze (Zhang et al., 2024) | Real-World | Multi-level haze, Homogeneous, Indoor | 5k |
| | HAZESPACE* (Islam et al., 2024) | 2D Synthetic | Multi-level haze, Vast range of scenes, Outdoor | 24×70k |
| | RESIDE* (Li et al., 2018) | 2D Synthetic | Multi-level haze, Indoor & Outdoor | 290k |
| | SYN-HAZE* | 2D Synthetic | Multi-level haze, Synthetic scenes, Include extremely dense haze, Indoor & Outdoor | 24×70k |
| Reflection | RRW (Zhu et al., 2024) | Real-World | Various scenes, Diverse glass and reflection types | 14.9k |
| | POLAR-RR (Lei et al., 2020) | Real-World | Polarization-based, Indoor | 0.8k |
| | RFC (Lei & Chen, 2021) | Real-World | Flash-induced reflections | 5k |
| | BDN (Yang et al., 2018) | 2D Synthetic | Linearly Blended, Public Image Sources | 50k |

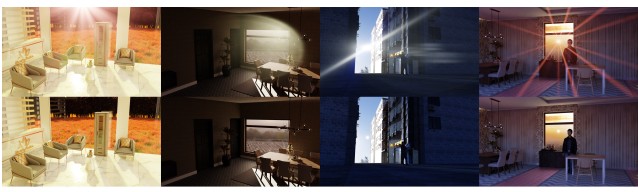
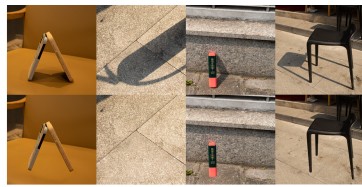

HALO                                                                       LR-SRD

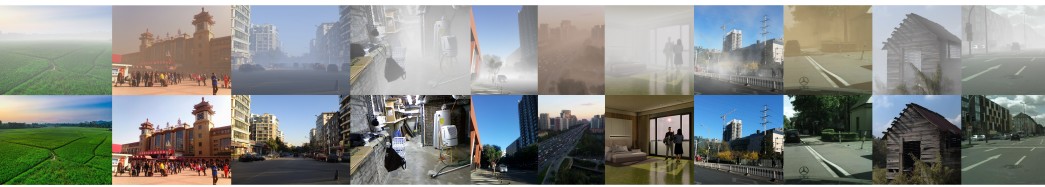

Synthetic Haze

Figure 2: Visualization of our curated data samples and synthetic haze by our method.

loss between the predicted noise and the ground truth noise, with a timestep-dependent weighting scheme to balance the contributions of different noise levels.

**Random Masking Strategy.** As established in the framework, a mask can be supplied as a condition to guide the denoising process toward a specific spatial region. However, most of the training sets do not contain the mask of effects. To ensure the model can robustly handle any user-provided mask shape, we adopt a random masking strategy. During training, following (Suvorov et al., 2022; Zheng et al., 2022) we synthesize a wide variety of binary masks $M$ by randomly combining geometric primitives like rectangles with free-form, stroke-like patterns that simulate user brush strokes. Afterwards, providing pairs $\{I_{input}, I_{gt}\}$, we generate the corresponding training supervision $I_{target}$ where the effect is removed only within the masked region via simply compositing $I_{input}$ and $I_{gt}$ with the mask, as shown in Equation 1. Note that the regions of effects in the input image are unavailable, hence the masks do not necessarily cover them. In this way the behaviors the model to learn is summarized as following:

- Region inside the mask w/ effects: remove effects based on the strength;
- Region inside the mask w/o effects: keep identical;
- Region outside the mask: keep identical.

Additionally, to make the supervision natural-looking, we blur the mask boundary via dilation and Gaussian blur. This strategy exposes the model to a vast distribution of possible mask shapes, enhancing its generalization capability for arbitrary user edits, and removing the sepcific effect regions.

**Removal Strength Control.** Beyond specifying *where* to remove an effect, OmniClear allows users to control *how much* of the effect is removed. This is achieved by training the model to interpret

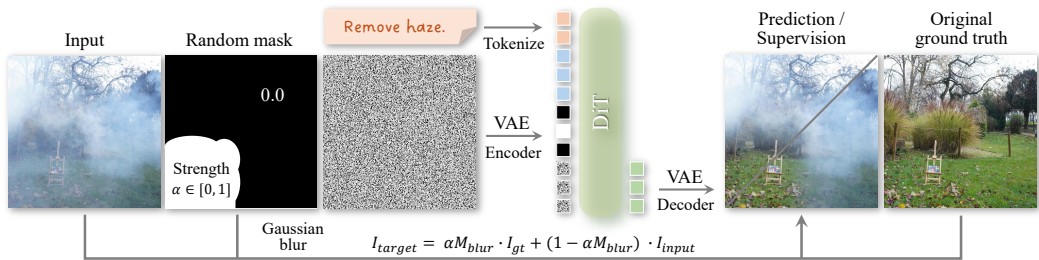

Figure 3: The architecture of OmniClear. During training, the mask is randomly synthesized along with a scalar strength, and the supervision is composed by the input image and the original ground truth via the mask and the strength.

continuous values in the conditional mask as an indicator of removal intensity. During the training process, for each sample, we uniformly sample a floating-point scalar value to represent "strength", denoted as $\alpha \in [0, 1]$. Instead of conditioning the model on a binary mask $M$, we provide a soft value mask $\alpha M$. The model thus learns to associate a mask value of 1.0 with complete removal, 0.0 with no change, and intermediate values with partial removal. On the other hand, along with the aforementioned blurred mask, the training target is generated by linearly interpolating between the clean ground truth ($I_{gt}$) and the input with effects ($I_{input}$) using the randomly sampled $\alpha$. Formally, the supervision during training is computed as following:

$$I_{target} = \alpha M_{blur} \cdot I_{gt} + (1 - \alpha M_{blur}) \cdot I_{input} \tag{1}$$

This joint strategy of conditioning on a soft mask while generating a correspondingly blended target enables the model to learn a continuous and intuitive mapping from the control signal to the desired degree of effect removal.

**Handling Undefined Effects.** Our framework also extends to zero-shot generalization on unseen soft effects through two complementary fine-tuning strategies. First, we randomly replace task-specific prompts with a generic prompt *"remove effects"*, encouraging the model to capture a shared notion of removal across tasks. Second, we introduce an auxiliary task using clean images: random masks are generated and overlaid with semi-transparent or opaque regions to synthesize degraded inputs, which are trained exclusively with the generic prompt. This prevents overfitting to predefined effect categories and compels the model to learn the broader concept of removing arbitrary occlusions, thereby enabling generalized restoration.

**Adding & Enhancing Effects.** We can easily invert the removal task to adding or enhancing effects by swapping the roles of the input and the target. Similarly, the adding or enhancing ability is controlled by the mask and strength given by users. We demonstrate this ability in Fig. 5.

## 4 EXPERIMENTS

### 4.1 BENCHMARKS AND BASELINES

**Benchmarks.** We evaluate OmniClear across four soft-effect tasks on widely used benchmarks. For *lens flare removal*, we adopt the Flare7K real-world test set (Dai et al., 2022). For *shadow removal*, we test on SRD (Qu et al., 2017), ISTD+ (Wang et al., 2018), and the high-resolution WSRD+ (Vasluianu et al., 2023). For *haze removal*, we use the SOTS and HSTS subsets of RE-SIDE (Li et al., 2018). For *reflection removal*, we employ $SIR^2$ (Wan et al., 2017) and the Nature test set (Li et al., 2020a). OmniClear is fine-tuned on the training splits of these datasets for domain adaptation. Evaluation uses standard full-reference metrics: PSNR and SSIM.

To assess real-world robustness, we collected 39 in-the-wild images containing haze, fog, flare, reflection, and shadow. As no ground truth is available, we report reference-free metrics (LIQE (Zhang et al., 2023), contrast gain (Wang et al., 2024b)), and a reference-based evaluation with Qwen2.5-VL-72B (Bai et al., 2025), a vision-language model instructed to judge the percentage of effect removal. We will further discuss these metrics in the supplementary material.

**Baselines.** We compare against both generalist and specialist methods. Generalist baselines include GPT-4o (Hurst et al., 2024), FLUX Kontext (Labs et al., 2025), Nano Banana (Google, 2025), and Seedream 4.0 (ByteDance, 2024). Specialist baselines cover:

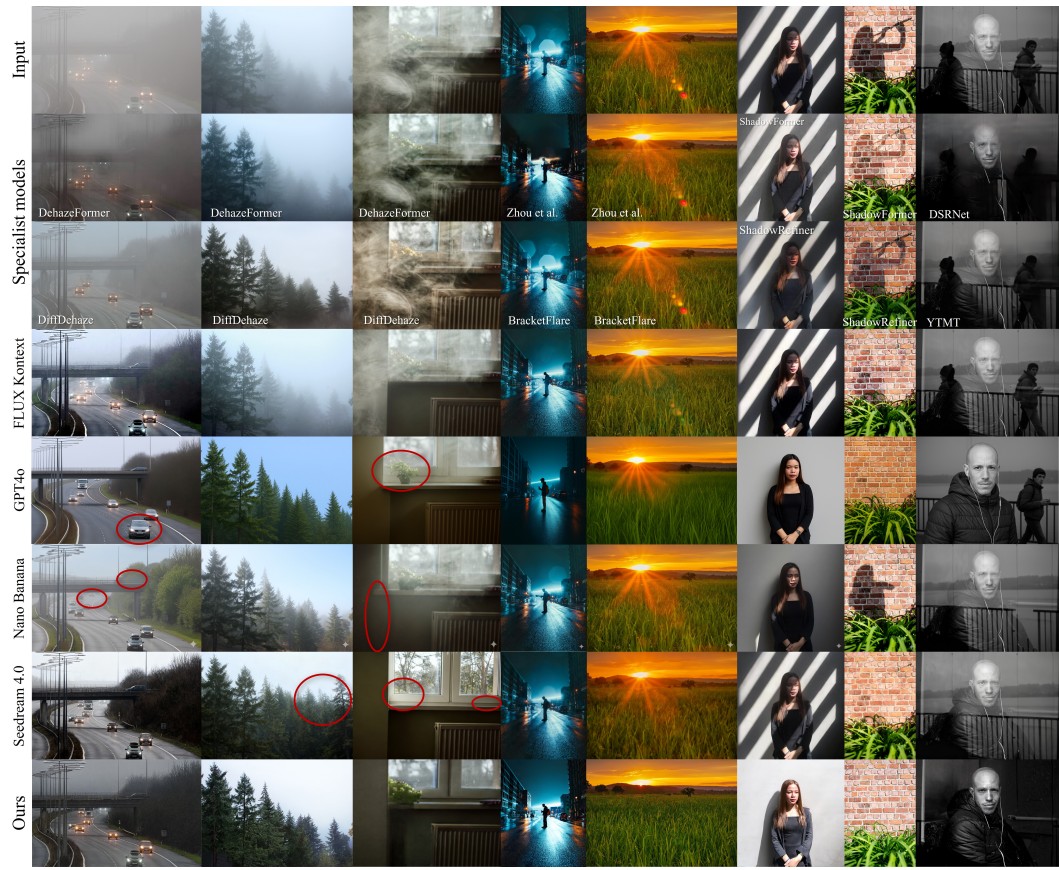

Figure 4: Comparisons with state-of-the-art specialist and generalist models on in-the-wild testing data. For effect removal, our method significantly outperforms these baselines. Moreover, generalist models fail to preserve the identity of background objects, some of the discrepancies are circled.

*Lens flare*: (Zhang et al., 2020; Zhou et al., 2023; Dai et al., 2022), BracketFlare (Dai et al., 2023), Difflare (Zhou et al., 2024);

*Dehazing*: DCP (He et al., 2010), AOD-Net (Li et al., 2017), GCANett (Chen et al., 2019), PSD (Chen et al., 2021), Dehazeformer (Song et al., 2023), MSF-Net (Zhu et al., 2021), UCL-Dehazet (Wang et al., 2024b), DiffDehaze (Wang et al., 2025);

*Shadow removal*: ShadowFormer (Guo et al., 2023a), ShadowRefiner (Dong et al., 2024), DCShadowNet (Jin et al., 2021), ShadowDiffusion (Guo et al., 2023b), StableShadowDiff (Xu et al., 2025);

*Reflection removal*: (Zhang et al., 2018), YTMT (Hu & Guo, 2021), DSRNet (Hu & Guo, 2023), PromptRR (Wang et al., 2024a), L-DiffER (Hong et al., 2024).

### 4.2 COMPARISONS WITH STATE-OF-THE-ART

**Qualitative Comparisons.** Fig. 4 visually compares OmniClear with state-of-the-art models on challenging in-the-wild images. Specialist models generalize poorly to out-of-domain data, often resulting in incomplete removal or new artifacts. Meanwhile, powerful generalist models like Nano Banana and FLUX Kontext suffer from instability and fail to preserve scene details, leading to significant content drift (highlighted by red circles). In contrast, OmniClear effectively removes a wide range of soft effects while remaining highly faithful to the original image content, producing clean and content-consistent results.

**Quantitative Comparisons.** To assess real-world generalization, we first conduct a comparison on a challenging in-the-wild test set using no-reference metrics, shown in Table 2. In this more difficult setting, OmniClear significantly outperforms both specialist and generalist baselines in terms of perceptual quality and removal efficacy, achieving the highest LIQE, Contrast gain, and QwenQA scores across nearly all tasks, which highlights its robust generalization. We then evaluate OmniClear against specialists on eight standard benchmarks using full-reference metrics (Table 3). The

Table 2: No-reference quantitative comparison on in-the-wild images for four SER tasks. We report results from multiple image quality assessment metrics.

| Haze | | | | Shadow | | | |
|---|---|---|---|---|---|---|---|
| **Method** | **LIQE↑** | **Contrast↑** | **QwenQA↑** | **Method** | **LIQE↑** | **Contrast↑** | **QwenQA↑** |
| Dehazeformer | 1.9999 | +0.74 | 0.0 | ShadowFormer | 3.3704 | +3.09 | 18.8 |
| DiffDehaze | 1.5624 | +0.03 | 9.1 | ShadowRefiner | 3.5179 | -2.30 | 26.3 |
| Flux Kontext | 2.2584 | +3.85 | 22.7 | Flux Kontext | 3.3184 | +0.73 | 36.3 |
| Nano Banana | 2.6864 | +0.26 | 27.3 | Nano Banana | 3.6399 | -4.93 | 35.0 |
| Seedream 4.0 | 2.1253 | +2.60 | 52.7 | Seedream 4.0 | 2.7640 | -3.58 | 36.3 |
| **Ours** | **2.8225** | **+5.57** | **60.0** | **Ours** | **3.7764** | **+3.61** | **65.0** |

| Lens Flares | | | | Reflections | | | |
|---|---|---|---|---|---|---|---|
| **Method** | **LIQE↑** | **Contrast↑** | **QwenQA↑** | **Method** | **LIQE↑** | **Contrast↑** | **QwenQA↑** |
| Uformer | 1.3832 | -4.39 | 30.9 | YTMT | 1.1187 | -2.30 | 14.4 |
| BracketFlare | 3.3377 | -10.72 | 13.6 | DSRNet | 1.6975 | -4.17 | 17.8 |
| Flux Kontext | 3.0574 | -0.31 | 62.7 | Flux Kontext | 1.7009 | +0.71 | 8.9 |
| Nano Banana | 3.0358 | -4.05 | 71.8 | Nano Banana | 2.0935 | -1.25 | 56.7 |
| Seedream 4.0 | 2.1643 | -4.41 | 73.6 | Seedream 4.0 | 1.6145 | -0.96 | 53.5 |
| **Ours** | **3.5186** | **+2.33** | **92.7** | **Ours** | **2.2257** | **+1.83** | **75.6** |

Table 3: Quantitative comparison with state-of-the-art methods across four soft effect removal tasks. We report PSNR (↑) and SSIM (↑) on eight benchmarks. Our unified model is compared against specialist methods in each respective category.

| Lens Flares | | | Haze | | | | |
|---|---|---|---|---|---|---|---|
| **Method** | Flare7k | | **Method** | HSTS | | SOTS | |
| | PSNR | SSIM | | PSNR | SSIM | PSNR | SSIM |
| Zhang et al. (2020) | 21.02 | 0.784 | DCP | 17.01 | 0.803 | 18.38 | 0.819 |
| Zhou et al. (2023) | 25.18 | 0.872 | AOD-Net | 19.68 | 0.835 | 20.08 | 0.861 |
| UNet (Dai et al., 2022) | 26.11 | 0.879 | GCANet | 21.37 | 0.874 | 21.66 | 0.867 |
| Restormer (Dai et al., 2022) | 26.28 | 0.883 | PSD | 19.37 | 0.824 | 20.49 | 0.844 |
| Uformer (Dai et al., 2022) | 26.98 | 0.890 | MSFNet | 31.03 | 0.931 | **30.07** | 0.939 |
| Difflare | 26.06 | **0.898** | UCL-Dehaze | 26.87 | 0.933 | 25.21 | 0.927 |
| **Ours** | **27.34** | 0.891 | **Ours** | **32.17** | **0.962** | 29.52 | **0.955** |

| Shadow | | | | | | Reflections | | | | |
|---|---|---|---|---|---|---|---|---|---|---|
| **Method** | WSRD+ | | ISTD+ | | SRD | | **Method** | SIR2 | | Nature20 | |
| | PSNR | SSIM | PSNR | SSIM | PSNR | SSIM | | PSNR | SSIM | PSNR | SSIM |
| ShadowFormer | 25.44 | 0.820 | 32.78 | 0.934 | 30.58 | 0.958 | Zhang et al. (2018) | 22.45 | 0.872 | 20.37 | 0.772 |
| ShadowRefiner | 26.04 | 0.827 | 31.03 | 0.928 | - | - | YTMT | 23.05 | 0.886 | 21.03 | 0.802 |
| DCShadowNet | 21.62 | 0.593 | 25.50 | 0.694 | - | - | DSRNet | 24.97 | 0.907 | 21.70 | 0.820 |
| ShadowDiffusion | - | - | 31.08 | 0.950 | 31.91 | 0.968 | PromptRR | 24.22 | 0.876 | 21.00 | 0.814 |
| StableShadowDiff | 26.26 | 0.827 | 35.19 | **0.970** | 33.63 | 0.968 | L-DiffER | 25.18 | 0.911 | 23.95 | **0.831** |
| **Ours** | **26.91** | **0.829** | **35.59** | 0.964 | **34.16** | **0.971** | **Ours** | **25.98** | 0.911 | **24.17** | 0.812 |

results show our unified model achieves state-of-the-art performance, consistently outperforming or matching specialist models by obtaining top scores across all four tasks, including the highest PSNR on multiple benchmarks.

### 4.3 DISCUSSIONS AND APPLICATIONS

**Ablation studies.** We conduct an ablation study to validate the effectiveness of our joint-task learning strategy. As shown in Table 4, we compare our full model, trained with Joint-Task Learning (JTL), against four same models trained independently using Single-Task Learning (STL). The results clearly indicate that the JTL model consistently outperforms the STL models across all four tasks on their respective benchmarks. This superiority suggests that by learning a unified representation from diverse soft effects, OmniClear develops a more robust and generalizable feature space that benefits all individual tasks.

**Strength control.** As illustrated in Figure 5(a), OmniClear provides fine-grained control over the intensity of the effect removal. Users can specify a continuous strength value, allowing for a smooth transition from partial reduction to complete removal of the artifact. This feature offers greater flexibility for users to achieve their desired level of restoration.

**Mask control.** OmniClear supports precise, localized editing through mask-based control, as shown in Figure 5(b). By providing a binary mask, users can designate specific spatial regions for effect

Table 4: Ablation study on training strategies. JTL (Joint-Task Learning) represents our full Omni-Clear, while STL (Single-Task Learning) denotes models trained separately for each task.

| Method | Lens Flares | | Haze | | Shadow | | Reflections | |
| | Flare7k | | HSTS | | ISTD+ | | SIR2-wild | |
| | PSNR ↑ | SSIM ↑ | PSNR ↑ | SSIM ↑ | PSNR ↑ | SSIM ↑ | PSNR ↑ | SSIM ↑ |
|---|---|---|---|---|---|---|---|---|
| STL | 27.18 | 0.890 | 31.91 | 0.963 | 35.43 | **0.963** | 26.40 | 0.876 |
| JTL | **27.34** | **0.891** | **32.17** | 0.962 | **35.59** | **0.964** | **27.44** | **0.918** |

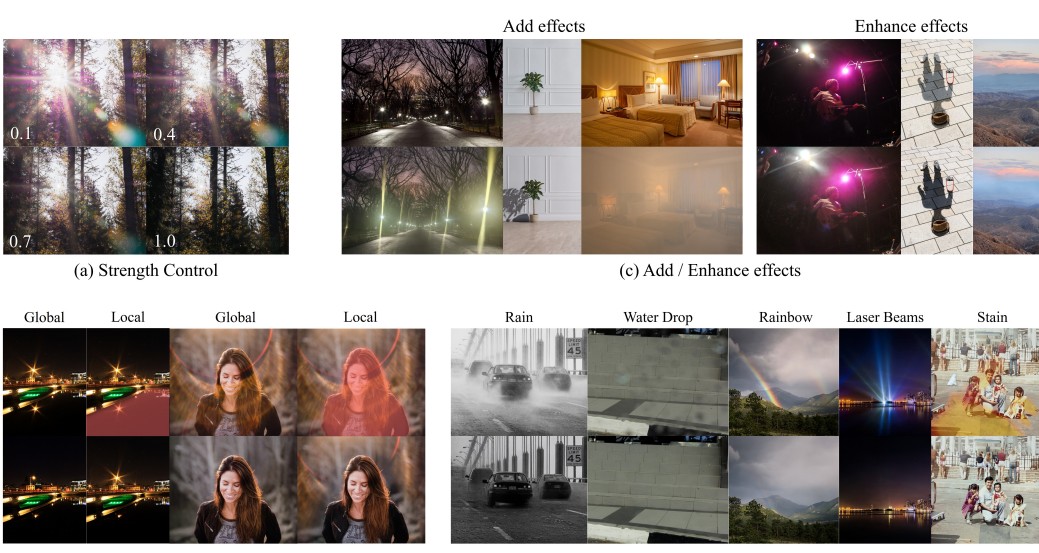

Figure 5: (a) Illustration of Strength Control for effect removal. (b) Illustration of Mask Control for accurate user regional editing. (c) Adding realistic effects to clean image, or enhance current effects for flexible editing purpose. (d) Zero-shot generalization ability on multiple unseen degradations like rain, stain, etc.

removal while leaving the rest of the image untouched. This allows for targeted and accurate edits tailored to user needs.

**Effects addition and enhancement.** Beyond removal, the OmniClear framework is also capable of generative tasks. As demonstrated in Figure 5(c), by inverting the process, our model can realistically add new soft effects to clean images or enhance existing ones. This versatility makes it a valuable tool for creative photo editing and data augmentation.

**Zero-shot removal.** OmniClear exhibits strong generalization capabilities to novel degradations not seen during training. As shown in Figure 5(d), the model can perform zero-shot removal of unseen artifacts such as rain and stains. This ability underscores the robustness of the learned features and the model's potential to handle a wide array of real-world image restoration challenges beyond its core training tasks.

**Reproducibility Statement** The portion of our method that relies on public datasets is reproducible, as our implementation is based on the open-source DiT codebase.

## 5 CONCLUSION AND LIMITATIONS

In this work, we introduced OmniClear, a unified foundation model for Soft Effects Removal (SER) that effectively handles diverse degradations including lens flare, haze, shadows, and reflections. Built upon a Diffusion Transformer trained on a large-scale pair-wise dataset, OmniClear overcomes the poor generalization of specialist models and the content inconsistency of generalist approaches. Extensive experiments demonstrate that our model achieves state-of-the-art performance on standard benchmarks and superior perceptual quality on in-the-wild images. Beyond high-quality removal, the framework provides fine-grained user controls, supports creative effect generation, and shows strong zero-shot capabilities on unseen degradations. Key limitations include its significant computational cost and the extensive resources required for training. Nevertheless, OmniClear represents a significant step towards a universal and controllable solution for high-fidelity image restoration.

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
