# OpenReview forum: "OmniClear: Soft Effects Removal from Images within a Versatile Model"
_ICLR.cc/2026/Conference — ICLR 2026 Conference Withdrawn Submission_

### Official Review · Reviewer_2ffc · 2025-10-19

**Soundness:** 2
**Presentation:** 3
**Contribution:** 3
**Rating:** 6
**Confidence:** 4

**Summary:**

This paper introduces OmniClear, a unified diffusion-transformer framework for Soft Effects Removal (SER), a generalized image restoration task encompassing diverse degradations. The model is trained on a large curated dataset and fine-tuned to support both global and mask-based controls, enabling continuous strength adjustment for effect removal and even effect addition. Built upon a latent diffusion architecture, OmniClear treats various degradations as semi-transparent occlusions and leverages text-prompt conditioning to perform task-agnostic restoration.

**Strengths:**

**Task Unification:** Tackles four traditionally separate restoration problems with a single framework.

**Strong Dataset Contribution:** The large-scale, curated training corpus (3.8 M pairs) fills gaps in existing domain-specific datasets and supports realistic learning.

**Comprehensive Evaluation:** Extensive experiments across multiple benchmarks and qualitative comparisons against both specialist and generalist baselines.

**Weaknesses:**

**Over-reliance on Data Scale:** Much of the gain appears to stem from large-scale data curation rather than methodological advancement; the paper lacks analysis separating model vs. data contributions.

**Limited Theoretical or Analytical Insight:** No formal discussion explains why treating diverse degradations as “semi-transparent occlusions” yields generalization benefits. The “shared essence” argument is intuitive but not rigorously justified.

**More Degradations:** Missing discussions of more types of degradations, e.g., Moire patterns (“P-bic: Ultra-high-definition image moiré patterns removal via patch bilateral compensation, MM 2024”), Blurring (“Unraveling Motion Uncertainty for Local Motion Deblurring, MM 2024”).

**Weak Comparison Baselines:** Missing comparisons with recent unified diffusion-based restoration or inpainting frameworks (e.g., UniRestore 2025, PromptIR 2023, diffusion-based editors with latent inpainting).

**Ablation Insufficiency:** The ablations are limited to single- vs. joint-task learning; no quantitative study isolates the effects of random masks, soft-strength conditioning, or dataset composition.

**Questions:**

Please refer to the weakness section.

---

### Official Review · Reviewer_1izE · 2025-10-31

**Soundness:** 2
**Presentation:** 3
**Contribution:** 2
**Rating:** 4
**Confidence:** 4

**Summary:**

The paper presents OmniClear, a unified framework for Soft Effects Removal (SER) — addressing multiple semi-transparent degradations such as haze, reflections, shadows, and lens flare. The method fine-tunes a diffusion transformer (DiT)–based inpainting model using a large curated dataset (~3.8M pairs) and provides controllable editing via pixel masks and strength modulation.
Results show state-of-the-art performance across four restoration domains and strong zero-shot generalization to unseen effects.

**Strengths:**

1. Clear and comprehensive writing: The paper is well organized and easy to follow. The methodology, dataset design, and experiments are described in significant detail, making the main ideas understandable.

2. Comprehensive data curation: The dataset section is meticulously designed, covering synthetic, real, and rendered data to achieve broad coverage.

3. Thorough evaluation: The experiments span eight benchmarks and in-the-wild images, with both full-reference and no-reference metrics. Comparisons to specialist and generalist baselines are convincing.

**Weaknesses:**

1. Limited Novelty of Framework:
The technical contribution of the core model is relatively modest.
OmniClear’s backbone is essentially a latent Diffusion Transformer trained on a large paired dataset — a combination similar to UniReal (CVPR 2025) and other diffusion-based restoration frameworks.
The innovations mainly lie in dataset scale and integration rather than new algorithmic or architectural insights. This makes the paper feel closer to a dataset-track or system-engineering contribution rather than a novel modeling paper.

2. Reproducibility & Transparency Issues:
The paper claims the implementation “is based on the open-source DiT codebase,” yet the supplementary material reportedly indicates that the actual backbone is an internal model not publicly available.
This discrepancy makes it difficult to assess the reproducibility or verify whether the reported performance stems from the proposed training pipeline or from proprietary model components.
Without clarity on the training setup (model size, optimizer, compute, pretraining data), it’s unclear whether OmniClear can be replicated or extended using existing open-source diffusion frameworks.

3. Lack of Open-Source Release:
The paper states that both the dataset and the model weights will not be released (at least not at submission).
Given that the main contribution hinges on large-scale data curation and fine-tuning, the absence of public availability significantly reduces its scientific value to the research community.
Future works will not be able to verify, benchmark, or build upon the results, limiting the paper’s long-term impact despite strong empirical performance

**Questions:**

1. Are there any plans to release a smaller or partial subset of the curated dataset or the pretrained OmniClear weights for benchmarking?

2. Synthetic Data Generation via OmniClear:
The paper claims strong zero-shot generalization and the ability to handle undefined effects.
Given this, do the authors plan to use the trained OmniClear itself to generate new paired SER data (e.g., simulate rare degradations such as rain, stains, or mixed effects)?
This would create a self-training or bootstrapping loop, enabling the community to expand data coverage where real pairs are scarce.
Releasing such synthetic data pairs could also be an effective compromise if the full real dataset cannot be shared due to licensing or resource constraints.

---

### Official Review · Reviewer_kjJc · 2025-10-31

**Soundness:** 2
**Presentation:** 4
**Contribution:** 1
**Rating:** 2
**Confidence:** 4

**Summary:**

The paper proposes OmniClear, a DiT-based framework for unified generalizable soft effects removal. It is trained on a large-scale dataset of 3.8M paired images, generated with high diversity. OmniClear provides a mechanism for controllable restoration through masks and a restoration strength, to satisfy different user preferences.

**Strengths:**

1. The paper proposes a large-scale high quality dataset for soft effects which is valuable to the community.
2. OmniClear achieves remarkable restoration performance with very high fidelity.
3. The paper is well written and easy to understand.

**Weaknesses:**

1. Limited Novelty: While the proposed approach is simple, it appears to be more of engineering than a novel methodology for soft effects removal. To me, the dataset seems to be the only novel contribution in the entire paper. While the mask-based training strategy allows for more controllability, there is no evidence supporting that it is responsible for the generalization performance of the method. In essence, the method leverages the powerful priors of DiT by fine-tuning it on a large-scale dataset for soft effects removal, which lacks novelty.
2. Lack of experimental validation: There is no experiment validating the effectiveness of the proposed dataset. Can the authors train OmniClear with only publicly available datasets to show the importance of the proposed dataset? Additionally, the authors need to provide an ablation to substantiate the claims in Lines 296-303.
3. Missing comparisons: While the authors focus on the importance of unified task learning, there are no comparisons with existing unified restoration approaches (eg. [1, 2, 3, 4]). Additionally, why are methods mentioned in Fig. 4 (DehazeFormer and DiffDehaze) missing from Table 3?

[1] Jiang, Yitong, et al. "Autodir: Automatic all-in-one image restoration with latent diffusion." European Conference on Computer Vision. Cham: Springer Nature Switzerland, 2024.

[2] Rajagopalan, Sudarshan, and Vishal M. Patel. "AWRaCLe: All-weather image restoration using visual in-context learning." Proceedings of the AAAI Conference on Artificial Intelligence. Vol. 39. No. 6. 2025.

[3] Tian, Xiangpeng, et al. "Degradation-Aware Feature Perturbation for All-in-One Image Restoration." Proceedings of the Computer Vision and Pattern Recognition Conference. 2025.

[4] Liu, Yuhao, et al. "Diff-plugin: Revitalizing details for diffusion-based low-level tasks." Proceedings of the IEEE/CVF Conference on Computer Vision and Pattern Recognition. 2024.

**Questions:**

See Weaknesses.

---

### Note · Authors · 2025-11-13

I have read and agree with the venue's withdrawal policy on behalf of myself and my co-authors.